

# Overnutrition in adolescents and its associated factors in Dale district schools in Ethiopia: a cross-sectional study

Beruk Berhanu Desalegn[1], Tona Zema Diddana[1], Alemneh Kabeta Daba[2] and Tagel Alemu Tafese[1]

[1] School of Nutrition, Food Science and Technology, College of Agriculture, Hawassa University, Hawassa, Sidama, Ethiopia

[2] School of Nursing, College of Medicine and Health Science, Hawassa University, Hawassa, Sidama, Ethiopia

## ABSTRACT

**Background.** Adolescence is the critical stage of an individual's growth and development that determines their nutritional status in the future. Adolescent overnutrition has become an increasing public health concern in developing countries like Ethiopia.

**Objective.** This study was designed to determine the magnitude and determinants of overnutrition among school-going adolescents in Dale District of Ethiopia.

**Methods.** An institution-based cross-sectional study was done between November and December 2020. A total of 333 school-going adolescents aged 10–19 years participated in this study. Socio-demographic, lifestyle, physical activity level, dietary energy intake, and height and weight data were collected. Body Mass Index for age Z-score (BAZ) was computed. Binary and multivariable logistic regression models were used to determine the association of outcome variable with explanatory variables, and results were reported using adjusted odds ratio (AOR) with 95% confidence interval.

**Results.** The magnitude of overnutrition was 7.2% (10.8% in the urban versus. 3.6% of rural schools). Overnutrition was positively associated with lack of sufficient play area within the school (AOR = 2.53, 95% CI [1.02–6.26]), being an urban resident (AOR = 3.05, 95% CI [1.12–8.29]), positive energy balance (AOR = 9.47, 95% CI [1.58–56.80]), consuming fast foods within a month before the survey date (AOR = 2.60, 95% CI [1.93–6.83]), having moderate (AOR = 9.28, 95% CI [6.70–71.63]) or low physical activity (PA) (AOR = 7.95, 95% CI [1.12–56.72]), and consuming snack within last one week before the survey date (AOR = 3.32, 95% CI [1.15–9.58]).

**Conclusion.** The magnitude of overnutrition among school-going adolescents was suboptimal. Sedentary lifestyles, excess calorie intake, having inadequate play areas within the school, and having snack and fast foods were determinants for overnutrition in the study area.

# INTRODUCTION

Overnutrition (overweight and obesity) is a globally increasing concern that affects a considerable proportion of children, adolescents, and adults (*FAO I, UNICEF, WFP and*

Corresponding authors
Beruk Berhanu Desalegn, beruk@hu.edu.et
Tagel Alemu Tafese, tagel@hu.edu.et

*WHO, 2019*). In 2016, an estimated 207 million adolescents were overnourished (*FAO I, UNICEF, WFP and WHO, 2019*). The ever-increasing burden of overnutrition has become an important public health problem in sub-Saharan Africa (SSA). A study that included 26 SSA countries' Demographic and Health Survey (DHS) data reported that 6.8% (10.7 million) of children in the region were affected by overnutrition (*Gebremedhin, 2015*). Ethiopia is an SSA country with a rising prevalence of overnutrition. For instance, a systematic review revealed that 11.3% of children and adolescents in Ethiopia were affected by overnutrition (*Gebre et al., 2018*). Another systematic review and meta-analysis revealed that one out of ten (11.39%) adolescents in Ethiopia are overnourished (*Diddana, 2021*). Additionally, district- and town-specific studies have reported that the magnitude of adolescent overnutrition varies from 9.78% to 20.54% in Ethiopia (*Alemu et al., 2014*; *Muluneh & Bishaw, 2018*; *Gali, Tamiru & Tamrat, 2017*; *Mekonnen, Tariku & Abebe, 2018*; *Anteneh et al., 2015*; *Dessalew, Mandesh & Semahegn, 2017*). The Ethiopian DHS report also witnessed that the trends of overnutrition showed increment from 0.4–0.6% in males and 2.4–3.4% in girls between 2011 and 2016 (*CSA and ICF, 2012*; *CSA and ICF, 2016*).

Although overnutrition is a result of the intricate interactions of biology, behavior, lifestyles, and the environment (*Askal et al., 2015*), the contribution of imbalance (positive) between energy intake and expenditure is profound (*Swinburn et al., 2011*). Overnutrition has a serious social, physical and psychological health impact on adolescents including self-image disturbance, low self-esteem, increased depression disorder, inadequate sleep, and reduced adult life expectancy (*Schwimmer, Burwinkle & Varni, 2003*; *Anderson et al., 2007*). It is also a risk factor for chronic morbidities, triggers an earlier onset of metabolic syndrome (*Anoop & Khurana, 2008*), and lowers the educational performance of adolescents (*Cairda et al., 2014*). From an economic perspectives, overnourished people have poorer job prospects; because they are perceived to be less performers; thus, they are less likely to be preferred by employers, less productive at work, and earn approximately 10% less than individuals with healthy body weight (*OECD/EU, 2016*). A systematic review revealed that absenteeism and presenteeism contribute to high indirect costs among overnourished servants compared with normal weight servants (*Goettler, Grosse & Sonntag, 2017*). Additionally, overnutrition has contributed to four million deaths globally (*Taylor et al., 2013*; *Global Disease Burden and The GBD 2015 Obesity Collaborators, 2017*).

The transition of childhood overnutrition into adulthood can be potentially halted during adolescence, making the period a second window of opportunity to correct growth- and development-related fallacies. Studies in Ethiopia have shown that socio-demographic and economic factors, schooling type, sedentary lifestyle, dietary practices that include an undiversified and unhealthy diet, poor nutrition knowledge, and attitude are associated with adolescent overnutrition. However, they are inconsistent across different areas (*Belay et al., 2022*; *Kedir et al., 2022*; *Worku et al., 2021*). Thus, the timely identification of context- and population-specific mediators and modification of risk factors are essential for designing appropriate interventions and preventing ever-increasing adolescent overnutrition problems. Moreover, the existing prevalence studies conducted in Ethiopia focused mainly on metropolitan towns; but not address adolescent overnutrition in rural and suburban settings. There is a lack of evidence regarding the effects of energy

balance (dietary energy intake and expenditure). Additionally, there is also a paucity of data on the current study area. Therefore, this study was designed to determine the magnitude and determinants of overnutrition among school-going adolescents in Dale District, Ethiopia.

## MATERIALS AND METHODS

### Study area and design
This study was conducted in selected high schools in the Dale District, Sidama regional state, Ethiopia. The district is 305 km away from Addis Ababa to the south of Ethiopia. It is approximately 1,200 m above sea level. Coffee, corn, barley, haricot bean, and beans are the most important cash crops grown in the district. Enset (*Ensete ventricosum*) and maize are staple crops. Fruits such as avocado and pineapple, and local varieties of cabbage (vegetable) are among the crops produced in the district. According to the District Education Office, 92 schools existed in 2017, of which, eight were high schools and preparatory schools. This school-based cross-sectional study was conducted between November and December 2020.

### Source and study populations
The source population of this study was adolescents (10–19 years old) attending rural and urban high schools (grade 9–12) in Dale District. Adolescents who were sampled from selected schools and recruited for interviews were the study population.

### Sample size determination
The sample size was calculated for the magnitude and factors associated with overnutrition. The sample size for the magnitude was determined using a single population proportion formula. The cumulative prevalence ($p = 15.6\%$) of overnutrition (overweight/obesity) was obtained from a previous study in Ethiopia (*Teshome, Singh & Moges, 2013*). The margin of error (0.05), critical value at the 95% confidence interval ($Z_{1-\alpha/2} = 1.96$), and design effect (DE) $= 1.5$, were used. A non-response rate of 10% was considered. Thus, the total sample size calculated for the first objective was 334. The second sample size was calculated by considering sex as a determining factor for overnutrition. For this, 80% power and a 95% confidence interval, with prevalence of control exposed (5.7%) and case exposure (20.1%) with an odds ratio of 5.14 were used (*Teshome, Singh & Moges, 2013*). A design effect of 1.5, and 10% non-response rate was considered to get a sample size of 234. Therefore, the largest sample size (334 adolescent) was used in this study. However, the final analysis considered 333 adolescents, as one participant left the interview before taking weight and height status.

### Operational definitions
Low physical activity is defined as a sedentary activity level that describes someone who has little to no exercise.

Moderate physical activity includes low-impact aerobic exercise, brisk walking or hiking, and recreational team sports (volleyball, soccer, etc.).

Vigorous physical activity levels include running or jogging, high-intensity aerobic classes, or competitive full-field sports.

Overweight is defined as an adolescent with a BMI for age z-score greater than +1SD and less than or equal to +2SD.

Obesity is defined as an adolescent with a BMI for age z-scores greater than +2SD.

Schools with playing areas are schools with adequate fields for their students to play football, volleyball, and handball games in relation to the proportion of students.

## Sampling techniques and procedures

A two-stage sampling technique was used to select representative samples. First, a list of eight high schools in Dale District was prepared; and stratified into urban (five high schools) and rural (four high schools). Then, two schools from each rural and urban setting were selected using simple random sampling. Second, the list of adolescents in each school was obtained from academic registrar records. Finally, the required sample was proportionally selected from each school by using a simple random sampling technique (lottery method) from the sampling frame.

## Inclusion and exclusion criteria

All adolescents who had attended education at the selected high schools during the data collection period were included. Adolescents with self-reported pregnancies during the data-collection period were excluded.

## Data collection instruments and procedures

The Ethiopia Demographic and Health Survey questionnaire (EDHS) was adapted to collect sociodemographic information (*CSA and ICF, 2016*). Individual adolescent health behaviors and lifestyle information were collected using mixed structured and semi-structured questionnaire.

Locally available utensils were standardized and used to estimate the amount of food items (weight equivalent) consumed within 24-hours before the survey date. An estimated 24-hours energy intake was determined from 24-hours quantitative dietary recall following a multiple-pass approach (*Gibson, 2019*). Food intake was converted into 24 h energy intake using the Ethiopian food composition (*EHNRI, 2000*) and NutriSurvey software.

The Physical Activity Level (PAL) was assessed using the Global Physical Activity Questionnaire (GPAQ). The tool consists of 16 questions about undertaking work, travel to and from places, recreational activities, time spent on activities, and sedentary behaviors. Before asking the questions, 10 min were given to each respondent to think all about the above-mentioned activities. Finally, physical activity level was classified into low, moderate, and high level based on responses (*World Health Organization, 2021*).

Harris-Benedict equation was used to compute basal energy expenditure (BEE) (*Harris & Benedict, 2018*). Total energy expenditure (TEE) was computed by multiplying BEE by physical activity level (PAL) (TEE = BEE*PAL) (*Schutz, Weinsier & Hunter, 2001*). Finally, an estimated energy (calorie) intake from 24-hour food recall and an individual's energy expenditure were compared. The energy balance was defined as being negative if dietary calorie intake was less than TEE, zero if it was equal to TEE, and positive if it was higher than TEE.

The weight and height measurement of the adolescents was taken. Shoes, bulky clothing, pins, and braids were removed before measuring these anthropometric indices. The UNICEF Seca digital weighing scale (Hamburg, Germany) was used to measure weight. Measurements were taken with respondents wearing light clothing, and weights were reported to the nearest 0.1 kg. Height was measured by a portable Stadiometer (Seca, Hamburg, Germany) and recorded to the nearest 0.1 cm. Using WHO AnthroPlus software 3.2.2. (*WHO, 2009*), the age and sex-specific body mass index for age (BAZ) was computed. BAZ was classified based on the WHO 2007 growth reference for adolescents. The nutritional status was finally defined as overnourished (BAZ >+1SD) and not overnourished (BAZ ≤+1SD) (*WHO, 2009*).

## Data quality control and assurance

For data quality assurance, first structured and semi-structured questionnaires were prepared in English language. Then, the questionnaire was translated into the Amharic language and back-translated to English to check consistency. Questionnaires, qualitative 24-hour dietary recall checklist, and anthropometric measurements were pretested by the 5% of the total sample size on the school other than the actual data was collected. The reliability and validity of the pretested questionere was checked. The reliability coefficient was calculated using Cronbach's alpha and it was greater than 0.7 for all tools. For each domain of the questionnaire, the validity of the questionnaire was checked by Pearson's correlation. A questionnaire with a correlation coefficient of less than 0.05 was declared valid.

The anthropometric indices measuring instruments were calibrated between each measurement. Duplicate measurement was taken by the same measurer to minimize measurement error. Two days of training was given for data collectors and supervisors on the data collection tools, questionnaire administration, anthropometric measurement procedures; and equipment calibration. Completeness and appropriate recording of the data was checked each day by the research team.

## Data management and statistical analysis

The data was coded and entered into a statistical package for social science (SPSS) software program for version 20 (*IBM Corp, 2011*). BAZ data was imported from WHO Anthro Plus into SPSS. Data was cleaned and checked for missing values. Kolmogorov–Smirnov test was used to check the normality of data. Multicollinearity was checked by the variance inflation factors (VIF) test and the variables with VIF less than 10 were considered not influential. The interaction of explanatory variables was checked by Pearson correlation analysis. Model fitness was assessed by the Hosmer-Lemeshow statistic test. The potential explanatory variables were socio-demographic data, lifestyles, individual health behavior, dietary habits, energy balance (intake and expenditure), and physical activity level. Two-step processes followed in regression analysis. First, bivariate logistic regression analysis was done followed by multiple logistic regression. A variable with a *p*-value of less than 0.2 in the bivariate analysis was considered a candidate variable for multivariate logistic regression. Potential candidate variables were entered into a multivariate logistic regression model to

adjust for confounders. At 95% confidence, variables with a probability value (*p*-value) less than 0.05 were declared statistically significant and determinants for overnutrition.

## Ethical approval and consent to participate

Ethical approval was obtained from the Hawassa University Institutional Review Board (IRB) with ethical approval number IRB/28/12. Informed written consent was obtained from students' families before starting data collection.

# RESULTS

## Socio-demographic characteristics

A total of 333 adolescents participated making a response rate of 99.7%. Nearly equal proportions of the adolescents were interviewed from urban (50.2%) and rural (49.8%) settings. About 88% and 11% of the participants were from public and private high schools, respectively. More than half (54.1%) of adolescents were in the age range of 15–17 years. About 69% of them were from a household with a monthly income of more than 1,000 Ethiopian Birr, and 59.2% were from farming households (Table 1).

## Lifestyle and related factors

In the 12 months before the survey date, the majority of the participants (87.1%) never consumed alcoholic beverages. Forty-two percent spent their free time on a sedentary lifestyle (watching television, playing mobile games, and computer screens). Regarding transportation, 30.9% of the adolescents had family transport of which 24% utilized frequently when they went to school. Having insufficient play areas in the school was reported by 50.2% of adolescents (Table 2).

## Dietary habits

The majority (89.5%) of the school adolescents had consumed snacks within one week before the survey date of which, 55.9% had consumed between 4–7 days within the this week. Over three-fourths (76.8%) of the school adolescents who participated in this study had a snack once per day. About 46.2% of the school adolescents included in this study had skipped breakfast within a week before the survey date. Over three-fourths (76.6%) of the adolescent students had consumed meals sometimes away from home (Table 3).

## Magnitude and determinants of overnutrition

In aggregate, the magnitude of overnutrition was 7.2% (95% CI [4%–10%]). It was 10.8% (95% CI [7.5%–14.1%]) *vs* 3.6% (95% CI [1.6%–5.6%]) in adolescents from rural *vs* urban schools, respectively. Concerning determinants, adolescents who had attended education in school with reported insufficient play area were two and half times more likely to be overnourished than those who attended schools with reported adequate play area (AOR = 2.53; 95% CI [1.02–6.26]). The odds of overnutrition was threefold higher (AOR = 3.05, 95% CI [1.12–8.29]) among adolescents from urban setting compared to those adolescents from rural area. Compared to the high physical activity groups, the odds of being overnourished was nearly eight times (AOR = 7.95, 95% CI [1.12–56.72]) and nine times (AOR = 9.28, 95 CI [6.70–71.63]) higher among adolescents engaged in low

**Table 1 Socio-demographic characteristics of school going adolescents in Dale district of Ethiopia, 2020, ($n = 333$).**

| Variables | | Frequency | Percent |
|---|---|---|---|
| Sex | Male | 169 | 50.8 |
| | Female | 164 | 49.2 |
| Age | Early adolescence (10–14) | 4 | 1.2 |
| | Middle adolescence (15–17) | 180 | 54.1 |
| | Late adolescence (18–19) | 149 | 44.7 |
| Relationship status with opposite sex | Single | 296 | 88.9 |
| | In a relation | 37 | 11.1 |
| Religion | Protestant | 230 | 69.1 |
| | Orthodox | 62 | 18.6 |
| | Others (Catholic and Muslim) | 41 | 12.3 |
| Ethnicity | Sidama | 309 | 92.8 |
| | Others (Wolaita, Gurage and Amhara) | 24 | 7.2 |
| School type | Public school | 294 | 88.3 |
| | Private school | 39 | 11.7 |
| Grade attended | Ninth | 152 | 45.6 |
| | Tenth | 153 | 45.9 |
| | Eleventh | 28 | 8.4 |
| Residence | Urban | 167 | 50.2 |
| | Rural | 166 | 49.8 |
| Head of the household | Father | 306 | 91.9 |
| | Mother | 19 | 5.7 |
| | Others (grand father and brother) | 8 | 2.4 |
| Family monthly income (ETB) | <500 | 30 | 9.0 |
| | 500–1,000 | 34 | 10.2 |
| | >1,000 | 232 | 69.7 |
| Occupation of the father | Farmer | 197 | 59.2 |
| | Government employee | 77 | 23.1 |
| | Self-employed (merchant) | 43 | 12.9 |
| | Laborer | 16 | 4.8 |
| Educational status of the father | No formal education | 107 | 32.1 |
| | Primary school | 104 | 31.2 |
| | Secondary school | 75 | 22.5 |
| | College level and above | 47 | 14.1 |
| Educational status of the mother | No formal education | 168 | 50.5 |
| | Primary school completed | 98 | 29.4 |
| | Secondary school and above | 67 | 20.1 |
| Family size | <5 person | 28 | 8.4 |
| | ≥5 person | 305 | 91.6 |
| Number of siblings in the home | Less than or equal to four | 171 | 51.4 |
| | Greater than four | 162 | 48.6 |

**Table 1** (*continued*)

| Variables | | Frequency | Percent |
|---|---|---|---|
| Number of cattle owned by the family | None | 57 | 17.1 |
| | 1–3 | 142 | 42.7 |
| | ≥4 | 134 | 40.2 |

**Table 2  Lifestyle characteristic of school going adolescents in Dale district of Ethiopia, 2020, (*n* = 333).**

| Variables | Frequency | Percent |
|---|---|---|
| Frequency of alcoholic beverages consumption within the past 12 months | | |
| Daily | 3 | 0.9 |
| Every 3 days | 11 | 3.3 |
| Weekly | 10 | 3.0 |
| Every 2 weeks or less | 19 | 5.7 |
| Never | 290 | 87.1 |
| Amount of one time alcoholic beverages consumed (*n* = 43) | | |
| 1–2 bottle | 21 | 6.3 |
| More than 2 bottle | 22 | 6.6 |
| Where spend free time | | |
| Watching TV/ playing computer or mobile games | 140 | 42.0 |
| Walking /sport | 193 | 58.0 |
| Sleep in afternoon | | |
| No | 291 | 87.4 |
| Yes | 42 | 12.6 |
| Number of days sleeping afternoon in a week (*n* = 42) | | |
| 1–2 | 27 | 8.1 |
| ≥3 | 15 | 4.5 |
| Number of sleeping hours in the afternoon per day (*n* = 42) | | |
| 0.5–1 | 27 | 8.1 |
| 1.5–4 | 15 | 4.5 |
| Families have transport | | |
| No | 230 | 69.1 |
| Yes | 103 | 30.9 |
| Frequently use family transport when go to school (*n* = 103) | | |
| No | 20 | 6.0 |
| Yes | 83 | 24.9 |
| Types of transport used (*n* = 103) | | |
| Bicycle | 11 | 3.3 |
| Motorbike | 60 | 18.0 |
| Others (automobile or Badjaj) | 12 | 3.6 |
| School have adequate play area | | |
| No | 167 | 50.2 |
| Yes | 166 | 49.8 |

or moderate physical activity level, respectively. The odds of being overnourished was 2.6 times (AOR = 2.6, 95% CI [1.93–6.8]) greater among adolescents who consumed fast

**Table 3 Dietary habits of the school-going adolescents in Dale district of Ethiopia, 2020, ($n = 333$).**

| Variables | Frequency | Percent |
|---|---|---|
| Consumed snack in the last one week | | |
| No | 35 | 10.5 |
| Yes | 298 | 89.5 |
| Number of days/week snack consumed ($n = 298$) | | |
| 1–3 | 112 | 33.6 |
| 4–7 | 186 | 55.9 |
| Number of snacks eaten per day in last one week ($n = 298$) | | |
| Once | 229 | 76.8 |
| 2–3 times | 69 | 23.2 |
| Skip breakfast | | |
| No | 179 | 53.8 |
| Yes | 154 | 46.2 |
| Number of breakfast skipped within last one week ($n = 154$) | | |
| 1–3 | 120 | 36.0 |
| 4–7 | 34 | 10.2 |
| Eat meal away from home | | |
| No | 78 | 23.4 |
| Yes | 255 | 76.6 |
| Number of meals eaten away from home within last one per week ($n = 255$) | | |
| 1–2 | 121 | 36.3 |
| 3–6 | 117 | 35.1 |
| ≥7 | 17 | 5.1 |

foods within a month before the survey date as compared with adolescents who never consumed. Being overnourished was more than nine times greater among adolescents with positive energy balance than those adolescents with zero or negative energy balance (AOR = 9.47, 95% CI [1.58–56.80]). Adolescents who consumed a snack at least one time in the week before the survey date were more than three (AOR = 3.32, 95% CI [1.15–9.58]) times more likely to be overnourished as compared to those who didn't have a snack (Table 4).

# DISCUSSION

Overnutrition (overweight and obesity) has a potentially serious impact on the physical and psychological health of adolescents. It lowers educational attainment and triggers an earlier onset of metabolic syndrome. The potential determinants vary across different settings. Exploring area-specific determinants is vital to designing appropriate interventions for preventing lifelong overnutrition and its consequences, particularly in Ethiopia where undernutrition and overnutrition are coexisting.

In aggregate, the magnitude of overnutrition among adolescent students was 7.2% (95% CI [4%–10%]). This finding is comparable with 9.78% reported in Addis Ababa (*Askal et al., 2015*) and 7.1% in Southwest Ethiopia (*Hassen, Gizaw & Belachew, 2017*). On

**Table 4  Determinants of overnutrition among school going adolescents in Dale district of Ethiopia, 2020, ($n = 333$).**

| Variables | Overnutrition | | Crude OR [95% CI] | Adjusted OR [95% CI] |
|---|---|---|---|---|
| | Yes (%) | No (%) | | |
| School have adequate play area | | | | |
| Yes | 5 (1.5) | 161 (48.4) | 1 | 1 |
| No | 19 (5.7) | 148 (44.4) | 4.13 [0.93, 4.96] | 2.53 [1.02, 6.26] |
| Residence | | | | |
| Rural | 6 (1.8) | 160 (48.1) | 1 | 1 |
| Urban | 18 (5.4) | 149 (44.7) | 3.22 [1.25, 8.33] | 3.05 [1.12, 8.29] |
| Energy balance | | | | |
| Negative or zero | 10 (3.0) | 283 (85.0) | 1 | 1 |
| Positive | 14 (4.2) | 26 (7.8) | 15.24 [6.16, 37.69] | 9.47 [1.58, 56.80] |
| Consumed fast foods one month before the survey date | | | | |
| Yes | 9 (2.7) | 68 (20.4) | 2.13 [1.81, 5.70] | 2.60 [1.93, 6.83] |
| No | 15 (4.5) | 241 (72.4) | 1 | 1 |
| Physical activity level (PAL) | | | | |
| High | 5 (1.5) | 257 (77.2) | 1 | 1 |
| Moderate | 2 (0.6) | 34 (10.2) | 48.54 [16.07, 146.67] | 9.28 [6.70, 71.63] |
| Low | 17 (5.1) | 18 (5.4) | 16.06 [3.33, 77.38] | 7.95 [1.12, 56.72] |
| Had snack within last one week | | | | |
| Yes | 18 (5.4) | 280 (84.1) | 3.22 [1.18, 8.75] | 3.32 [1.15, 9.58] |
| No | 6 (1.8) | 29 (8.7) | 1 | 1 |

**Notes.**
1, reference group; OR, Odds ratio; 95% CI, 95% Confidence interval.

the contrary, it is lower than a study reported in Gondar (13.8%) (*Bekele Sorrie, Yesuf & GebreGyorgis GebreMichael, 2017*), Diredawa (20.54%) (*Dessalew, Mandesh & Semahegn, 2017*), Southern Ethiopia (11.20%) (*Ajema Berbada et al., 2017*), Hawassa (12.82%) (*Teshome, Singh & Moges, 2013*), Bahirdar (16.46%, 12.5%) (*Anteneh et al., 2015*; *Worku et al., 2021*), and Addis Ababa (11.25%, 17.01%, 14.95%, 21.20%) (*Shegaze et al., 2016*; *Moges et al., 2018*; *Girmay & Hassen, 2018*; *Fitsum et al., 2021*). The value is higher than a study reported in an urban area of Ethiopia using 2011 Ethiopian Demographic and Health Survey data (*Mekonnen & Bogale, 2017*). These differences could be because of the socio-economic, environmental, dietary, residential, and cultural disparities and study period. Furthermore, nearly half of the school-going adolescents in this study were from rural and suburban residents, while almost all of the other studies done in Ethiopia were done in metropolitan towns. Rural and suburban populations are less likely to consume highly processed energy-dense and fast foods, less likely to utilize public and private transport, and spend more on labor-intensive work. These might result in less energy being stored in the tissues, which would reduce body weight gain. Additionally, agroecology and food production systems have an impact on calorie intake and food consumption. Fruits and vegetables like green kale are produced and consumed in large quantities in the current study area. Kocho, a very fibrous snack derived from enset (*Ensete ventricosum*), is a common dish in the region. This may slow down the pace at which calories are absorbed

and heighten feelings of fullness. Because of their low energy density, high fiber content, and satiety value, fruits and vegetables have been encouraged for the prevention of juvenile and adolescent obesity, according to the American Dietetic Association (*American Dietetic Association, 2006*).

The magnitude of overnutrition was higher in adolescents from urban schools (10.8%) than from rural schools (3.6%), which is in line with what was found in EDHS 2019, indicating that a higher percentage of adolescents in the urban setting is at risk of chronic non-communicable diseases (*Mathur & Pillai, 2019*).

Concerning the determinants, adolescents attending schools having inadequate play areas were 2.53 times more likely to be overnourished. This finding is in line with a study reported in Ethiopia and other countries (*Moges et al., 2018*; *Wall et al., 2012*; *Qian, Gaduh & Nayga, 2015*; *James et al., 2011*). This could be because having an adequate play area gives more opportunity for exercising and being physically active.

Adolescents from urban schools were three times more likely to be overnourished compared to adolescent students from rural schools. The possible explanation for this is that urban dwellers spend more time on sedentary activities like spending time on watching television or video, computer use, reading, and sitting at a desk or using transport compared to the rural. Consequently, prolonged time spent on a sedentary lifestyle limits energy expenditure and displaces light-intensity physical activities leading to weight gain over time. On the other hand, transportation modalities and dietary habits of urban and rural settings are different. This finding is in agreement with other studies (*Yeshaw et al., 2020*; *Ahmed & Tomas, 2015*; *Beyen, Gebregergs & Yesuf, 2013*).

Changes in body mass are mostly caused by disturbances in the energy balance. This is witnessed by this study in which, being overnourished is 9.47 times higher among adolescents with positive energy balance. This is because when energy intake exceeds expenditure, the excess energy is deposited as fat in the body tissue (*Anderson et al., 2015*). Scientific evidence also supports this relation by which a positive energy balance wherein energy intake exceeds expenditure causes weight gain, with 60–80% of the resulting weight gain being attributable to body fat (*Hill James & Commerford, 1996*).

In this study, adolescents having low and medium physical activity levels were 7.95 and 9.28 times more likely to be over-nourished compared to high physical activity level. Similar results were reported by other scholars from Bahirdar (*Mekonnen, Tariku & Abebe, 2018*), Southwest Ethiopia (*Gali, Tamiru & Tamrat, 2017*) and Addis Ababa town (*Dereje, Yirgu & Chichiabellu, 2018*). This could be because of energy expenditure or burning extra calories is less in less physically active adolescents as physical activity is a predisposing factor to burn excess energy accumulation in the body.

The odd of being overnourished was 3.32 times higher among adolescents who had snack compared to those who didn't consume. This finding is in line with other studies conducted in Hawassa (*Teshome, Singh & Moges, 2013*), and Gondar (*Beyen, Gebregergs & Yesuf, 2013*). This could be because eating snacks could provide extra daily energy intake.

Fast food intake had a significant association with overnutrition in the present study. This result is consistent with the studies from Hawassa (*Teshome, Singh & Moges, 2013*), Bahirdar (*Mekonnen, Tariku & Abebe, 2018*), and Addis Ababa (*Gebregergs, Yesuf & Beyen,*

*2013*). Another study from Lebanon (*Shegaze et al., 2016*) also found a positive association between fast food consumption and overnutrition. This might be related to the higher energy content of fast foods. However, contradicting results have been reported from Jimma (*Gali, Tamiru & Tamrat, 2017*) in which fast food consumption frequency was not associated with the incidence of overnutrition among adolescents. This could be due to individual variation in physical activity level, additional dietary habits, and serving size and amount of calories contributed by snacking.

## CONCLUSION

The magnitude of overnutrition among school-going adolescents was 7.2%. It was threefold higher among school adolescents in urban (10.8%) compared to school adolescents in rural settings (3.6%). Lack of adequate play area in the school, being an urban resident, positive energy balance, fast food consumption, and low and moderate physical activity level, and frequent eating of snacks were determinants for overnutrition in the study area. Therefore, future efforts to tackle overnutrition among school adolescents should consider place of residence, playing area in schools, encouragement of regular physical activity and promotion of health, and diversified diet consumption by minimizing the consumption of fast foods and encouraging consumption of snacks with adequate number and portion size, through school-based nutrition education program. Further research should be done using biomarkers for identifying the NCDs among adolescents coupled with school-based nutrition-specific and sensitive interventions.

## ACKNOWLEDGEMENTS

The authors would like to thank and acknowledge Mr. Tadesse Kanko, who passed away in a sudden car accident while we initiated this manuscript for the publication process. He had a leading role in the conceptualization, data collection, and writing the thesis, from which this manuscript was extracted. The authors also gratefully thank all study participants, data collectors and supervisors involved in this study.

### Funding
The authors received no funding for this work.

### Competing Interests
The authors declare there are no competing interests.

### Author Contributions
- Beruk Berhanu Desalegn conceived and designed the experiments, performed the experiments, analyzed the data, authored or reviewed drafts of the article, editorial role, journal searching and manuscript preparation, and approved the final draft.

- Tona Zema Diddana conceived and designed the experiments, analyzed the data, prepared figures and/or tables, authored or reviewed drafts of the article, and approved the final draft.
- Alemneh Kabeta Daba conceived and designed the experiments, authored or reviewed drafts of the article, and approved the final draft.
- Tagel Alemu Tafese analyzed the data, authored or reviewed drafts of the article, editorial role, journal searching and manuscript preparation, and approved the final draft.

## Human Ethics

The following information was supplied relating to ethical approvals (*i.e.*, approving body and any reference numbers):

Hawassa University Institutional Review Board granted Ethical approval to carry out the study (Ethical Application Ref: RB/28/12)

## Data Availability

The raw data is available in the Supplementary File.

## Supplemental Information

Supplemental information for this article can be found online at http://dx.doi.org/10.7717/peerj.16229#supplemental-information.

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
