# Peer review of "Overnutrition in adolescents and its associated factors in Dale district schools in Ethiopia: a cross-sectional study"

_PeerJ, doi:10.7717/peerj.16229_

## Round 0.1 · original submission · Minor Revisions

Please authors, kindly find the feedback of reviewers and note carefully their annotated comments.

Please go through all annotated comments first, then assemble all comments to one pdf, prior to addressing them, so that one can know where there is overlap/contrast of one reviewer's comment over the other(s)

Do due diligence in revising your manuscript and provide as much detail as possible, not only in the revised manuscript but also in your reply to each reviewer's comments.

Very best wishes

·

Basic reporting

Suggestions for Authors
Title: Overnutrition in adolescents and its causes in Dale District schools in Ethiopia: a cross-sectional study

1. Keywords
School appears to be too general could be replaced by Body Mass Index?
The first paragraph needs to be referenced.
2. Material and methods
Data collection instruments and procedures.
What other literature are you referring to in lines 103-104? It will be good to refer to the specific literature and reference them appropriately.
3. Discussion
Line 238-240, “The magnitude is three folds higher in adolescents from urban school as compared to rural indicating that higher percentage of adolescents in the urban setting are at risk of chronic non-communicable diseases”.
This statement appears so emphatic. Provide a linkage of NCDs and overnutrition and or a reference.
Line 237-243, When disaggregating residentially?
Line 248-255. Respondents from urban residents.
The differences between these two (Line 237-243 and Line 248-255) are not clear. Clearly distinguish between the schools and the adolescents or the two can be merged.
Table 1
Some variables have a total exceeding the sample size.
e.g., Religion has 274 exceeding the total of 333.
Number of cattle owned by the family has 212 which is less the total of 333.
4. IT will require an extensive English Language edit as some expressions and words may not be contextually appropriate.
e.g., Line 10, 104, 179, 185-190, 195-196.

Good manuscript overall.

Experimental design

None.

Validity of the findings

None.

Additional comments

None.

·

Basic reporting

Clear and precise English was used throughout the work with sufficient references. However, the introduction lacks adequate structure from overnutrition, the causes, effects and remedies. Where does overnutrition emanate from? Are parents and childminders fundamental to this from birth to the adolescent years? Were they given the right proportion of all the classes of food? This information was not captured in the introduction.
I noticed grammatical errors in line 50 through line 52, the sentences need to be rewritten. In lines 56 to 59, the authors stated that overnourished people tend to have poorer prospects. Can they explain why this is so?

Experimental design

My field of specialization is not in line with this kind of experimental design but the results from the data were sufficient and properly arranged. The aim of the work is in line with the scope of the journal and tries to fill a knowledge gap. The research questions were very clear, important and needs to be addressed to avoid future problems. The number of participants engaged in the study was adequate and their feedbacks were encouraging.

Validity of the findings

Reading this article from a layman’s point of view, the authors covered a wide range of experiments from Socio-demographic characteristics to lifestyle characteristics to dietary habits of school-going adolescents and the determinants of overnutrition generally. It will be critical for a nutrition expert to look at the tables, the research questions and the feedback to ascertain the validity of the findings.

Additional comments

Overall, the research was extensive and covered a wide range of causes of overnutrition in school-going adolescents. As suggested earlier, more information should be added to the introduction to give a better understanding of the effects, causes and remedies of overnutrition in adolescents.

Reviewer 3 ·

Basic reporting

.

Experimental design

.

Validity of the findings

.

Additional comments

Need further improvement. See attachment.

Annotated reviews are not available for download in order to protect the identity of reviewers who chose to remain anonymous.

---

## Round 0.2 · Minor Revisions

Reviewers have considered your work to merit publication. However, please English language needs to be improved.
Kindly consider the following:

(i) have a colleague who is proficient in English and familiar with the subject matter review the manuscript; OR
(ii) contact a professional editing service to review the manuscript.

Look forward to your revised manuscript. Thank you

**Language Note:** The Academic Editor has identified that the English language must be improved. PeerJ can provide language editing services - please contact us at copyediting@peerj.com for pricing (be sure to provide your manuscript number and title). Alternatively, you should make your own arrangements to improve the language quality and provide details in your response letter. – PeerJ Staff

·

Basic reporting

The quality of the manuscript has been considerably improved. Thank you.

Experimental design

no comment

Validity of the findings

no comment

Additional comments

no comment

·

Basic reporting

Clear and precise English was used throughout the work, with sufficient references. The introduction is now better constructed and flows accordingly for better understanding. The authors have explained why overnourished people tend to have poorer prospects

Experimental design

The results from the data were sufficient and properly arranged, and the feedback was adequate.

Validity of the findings

The research questions and the feedback to ascertain the validity of the findings were addressed properly.

Additional comments

No comment.

---

## Round 0.3 · accepted · Accept

Thank you authors for the English improvement, which has further elevated the quality of the revised manuscript. It is acceptable for publication. Thank you for finding PeerJ as your journal of choice, and I look forward to your future scholarly contributions. Congratulations.